# Understanding EU Fisheries Management Dynamics by Engaging Stakeholders through Online Group Model-Building

Erda Gercek [1,*], Monica Gambino [2] and Loretta Malvarosa [2]

1 Graduate School of Business, Koç University, Istanbul 34450, Turkey
2 NISEA, Fishery and Aquaculture Economic Research, 84100 Salerno, Italy
* Correspondence: egercek@ku.edu.tr

**Abstract:** The Common Fisheries Policy (CFP) has a challenging mandate to find the right policy mix to simultaneously achieve all three aspects of sustainability: economic, social, and environmental. But development and implementation of an effective and sustainable fisheries management policy has been a challenge all over the world. The evidence of this failure is found in the continuous decline in fish stocks. Faced with the difficulty in fulfilling this mandate, the European Commission has increasingly been embracing fisheries stakeholders' involvement. Stakeholder involvement in policy development and implementation is important because it tries to bring the relevant interested parties together, understanding and paying attention to what is important to each and every stakeholder, identifying the individual and common issues. This process in turn can foster connections, trust, confidence, and buy-in, and commitment for the implementation of the policy. This research describes a group model-building (GMB) approach using system dynamic methodology as a participatory model building tool, enabling stakeholders to become deeply involved in the identification and modelling of the complex issues faced by the EU fisheries. Given the geographical diversity of the stakeholders, GMB was applied online, both synchronously and asynchronously, providing participants time to carefully reflect on key variables, their relationships, and the behaviour of the overall system. The study demonstrated the need and relevance of an adequate engagement of the stakeholders, with online stakeholder consultation proving an effective method of engagement. Hence, the study is very relevant for both scientists and managers. The GMB process meant the final model evolved significantly from the initial one offered, which pointed to active involvement in and progressive learning from the modelling process itself, as the methodology argues. Two quantitative stock-flow models using actual numbers were built not only to aid the GMB process but to depict how all three aspects of sustainability could actually be met with the right set of policies that consider feedback loops and inherent trade-offs.

**Keywords:** sustainability; stakeholders; group model building; causal loop diagrams

## 1. Introduction

Development and implementation of an effective and sustainable fisheries management policy has been a challenge all over the world. One of the key messages from the FAO report is the need for active fisheries management as the agency declared the state of marine fishery resource has continued to deteriorate [1]. This research tries to address the fisheries management challenges in a novel way by applying a system dynamics methodology (SD) qualitative group model-building tool online with the participation of relevant stakeholders. The qualitative process first endeavours to depict the state of the system and then generate policy options. Two quantitative models concurrently constructed envisage to ascertain under what policy mix the three pillars of sustainability would make CFP targeting achievable.

This study picked up where Dudley [2] left off in his formative work. The novelty of his research was putting the Schaefer biomass model into SD format. His working

model demonstrated how unmanaged fisheries result in serious fish biomass reduction and how effective management could address this sustainability challenge. In his research, having answered the question of how a complex and uncertain system like fisheries could be managed with sound policy decision making tools, Dudley [2] raised several other issues for future researchers to pursue. This article intends to take his research further by answering two of his questions: "Which management regimes best encourage cooperation between managers and users, and among users? [....] In a more general sense, can transparent system dynamics models be used more widely to encourage discussion of complex fishery management issues in an open and constructive atmosphere?" [2] (p. 25). In addition to answering his questions, this study built on his conceptual fisheries model, which formed the basis for the causal loop diagram (CLD) produced by participating stakeholders and generated two stock-flow diagrams with actual numbers for two respective EU marine regions to gauge not one, as was the case with Dudley's research, but all three aspects of sustainability.

Given the vast number of fishery stakeholders, their different needs and claims, a participatory process was envisaged to generate a common understanding of the challenges and develop a better decision-making platform. Among many participatory modelling tools, SD group model-building (GMB) of a causal loop diagram was preferred due to its best fit for what was being targeted. Through a three-stage process, mental models of the individual stakeholders were elicited to a build causal loop diagram to not only depict the state of the system but to generate policy options, which was reinforced by the accompanying quantitative stock-flow models and their equations (shared in Appendix A). The results section explains these policy options through scenario analysis, after the details of the evolution of the group-constructed CLD were laid out. The discussion part deep dives into the novelty of this research and compares and contrasts it with a limited number of similar studies.

This research contributes to the literature in several ways. First, an SD tool of causal loop GMB was applied online synchronously and asynchronously for the first time to European fisheries with input from a carefully selected diverse set of stakeholders. This heterogonous group of stakeholders that belonged to various different organisations across different countries made an online process inevitable along with constraints generated by the pandemic. This should pave the way for other researchers to leverage this method to better calibrate both the number of researchers and participants necessary to facilitate such method. Second, modelling of fisheries management issues was aided by the construction of quantitative models with actual numbers, not through simulations, to ascertain whether the CFP's mandate of meeting all three aspects of sustainability is achievable. Finally, these quantitative models are made available online to policymakers to test the outcomes of different policy options before they are implemented to avoid unintended consequences.

Systems approach argues that managing a fishery means changing its purpose from catch maximisation towards achieving a more sustainable outcome for both the fish stock and the fishers, as Meadows [3] argued. Complex systems are characterized by strong (usually non-linear) interactions between various parts, and feedback loops make it difficult to distinguish cause from effect due to significant time and space discontinuities, thresholds, and limits. Thus, before interfering in the system with various management measures such as limitation on the number of licenses, catch quotas, landing obligations, spatial and temporal restrictions, and restriction on gear characteristics and technical features of the fishing vessels, a true and holistic understanding is necessary since such interventions increases the complexity of the system. As new connections and feedback loops emerge, the behaviour of the system can evolve in ways not foreseen by managers.

Fisheries exhibit most of a complex system's characteristics, making fisheries management a challenging process. Failures of fishery management have been widely studied. Like in most management approaches, most decisions tend to concentrate on short term results, often leading to unintended consequences as delayed feedback loops are seldom considered. Adaptive behaviour is regularly overlooked when considering restrictions in

the case; for example, in response of fishers to shorter fishing periods by increasing catching gear efficiency. In essence, complexity of the system begets more complexity, paving the way for new institutions leading to ever more regulation, more adaptive behaviour, and the vicious cycle intensifies.

Sustainability in complex systems is harder to achieve and the starting point is to understand the systems approach to sustainability that is depicted in Figure 1. This basic system dynamics stock-flow (SF) notation help operationalise World Bank economist Herman Daly's [4] approach to sustainability. Daly [4] stated that a sustainable world would not use renewable resources (forests, soils, waters, fish, and game) faster than they were replenished; it would not use non-renewable resources (fossil fuels, mineral ores) faster than renewable substitutes could be found for them; it wouldn't release pollutants faster than the earth could process to make them harmless.

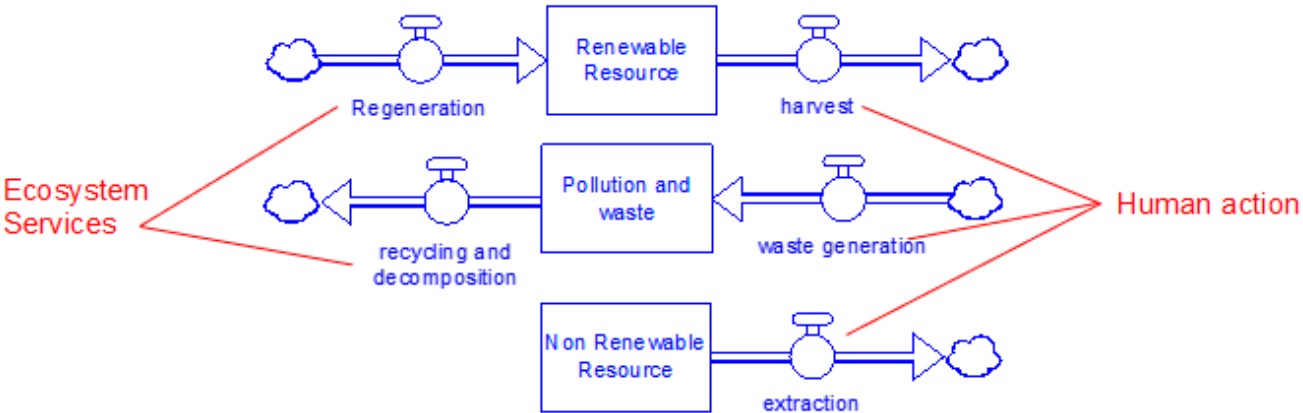

**Figure 1.** Adopted from [5] produced by Stella Architecture.

In Figure 1 rectangles represent stocks; pipes and valves represent the flows. The right side of these diagrams are due to human decisions and actions and the left side is just like a service of the earth's ecosystem to benefit all. In the first two stock-flow representations, to the extent that regeneration (e.g., reforesting) equals harvest (e.g., deforesting), and waste generation (e.g., $CO_2$) equals recycling and decomposition (e.g., sequestration), respectively, then there is equilibrium and stocks remain at initial levels. As far as the third one is concerned there is no eco-service to replenish those resources but only renewable alternatives to replace them as they disappear. In fact, there are no limitless resources in a finite world, except for human ingenuity to address these issues.

Systems view of sustainability is significantly different than the three-pillars representation of sustainability or sustainable development, which has been variously attributed to the Brundtland Report, Agenda 21, and the 2002 World Summit on Sustainable Development. These pillars give the wrong impression that they are independent from each other and they collectively "carry" the load of sustainability, completely disregarding the interconnectedness, interactions, and the dynamic behaviour of all three components and sustainability's emergent property. In such a system, management, as well as decision-makers and actors, are endogenous and hence should co-evolve with the very system they are mandated to manage to address sustainability challenges more effectively. Management, in essence, is an art of basically "designing and redesigning the system of feedback loops in such a way as to achieve the most satisfactory series of levels" [6] (p. 540).

Complex, dynamic and uncertain systems such as ecology and economy are inherently difficult to model to gauge sustainability and hence call for a different approach. Involving stakeholders across many disciplines, with diverse and sometimes conflicting knowledge and incentives, alongside experts and politicians can help better define the problem and generate solutions. Participatory model-building (PMB) is increasingly being recognised as a robust method for collaborative decision-making for complex and uncertain systems. Voinov et al. [7] (p. 233) defined PMB in a very concise way as a "purposeful learning

process for action" that drove out the "implicit and explicit knowledge of stakeholders to create formalized and shared representations of reality".

## 2. Materials and Methods

This section first describes the choice of the participatory modelling tool, then discusses the process for identifying stakeholders before deep diving into the specifics of the novel methodological approach undertaken.

### 2.1. System Dynamics as Participatory Model Building Tool

There are various PMB tools available to modellers and stakeholders. Voinov et al. [7] conducted an extensive survey where users identified system dynamics as familiar in interviews and named it as the most preferred method (26%), followed by CLD, which is also a system dynamics tool that is applied to this research. SD's capability scores relative to other PMB tools were high in temporal representation, qualitative and quantitative forecasting, and feedback loops; as well as handling uncertainty and medium in transparency, ease of communication, and modification. The key discriminating factor of SD is the feedback; the fact that X affects Y and then Y affects X sometimes through a chain of causes and effects (SDS, n.d.). The "system" in system dynamics is defined as a set of interrelated elements organised to serve a function or goal. The behaviour arises out of the system from the interconnectedness of complex patterns with stocks, flows, and feedback loops; where non-linear processes and delays are intrinsic to the system; and information flows are different than physical flows. Modelling starts with identification of a problem, an inquiry motivated by undesirable system behaviour. First the problem needs to be understood. After conceptually describing the system, a hypothesis (theory) is generated for how the system is creating the troubling behaviour [8]. The single most important thing about SD is that the modeller gains in-depth knowledge on how a structure generates a dynamic behaviour. A CLD is one such qualitative SD tool to map out the cause-and-effect relationships that are assumed to give rise to a certain problem or pattern of behaviour. They help analyse why certain problems may be occurring and why certain solutions may or may not work. Feedback loops emerge under close scrutiny. Understanding the nature of the feedbacks in a system is essential to comprehend how a system is likely to behave. A positive feedback loop reinforces itself over time. The theorem is that depending on whether it is reinforcing itself positively or negatively a positive feedback loop generates exponential growth or exponential decline. A negative feedback loop, on the other hand, behaves in a goal-seeking fashion over time, regardless of what side of the desired level it is on in the beginning [9].

Group model-building (GMB) is an SD model building process where the members of an organisation, trying to address a problem, become deeply involved in the construction of the model. The essential idea is that as the modelling process unfolds the problem and eventually ways to deal with the problem become clearer for everyone.

This interest in GMB was due mostly to the benefits summarised by Vennix [10] as (a) capturing the knowledge tacit in the mental models of the participants; (b) enhancing the participants' learning process as most of the learning takes place in the process of building a model rather than when the model is finalised (modelling as learning); (c) increasing the validity of the model; (d) involving the client increases the chance of implementation. Online GMB has the added advantage of scalability in terms of number of participants, regardless of their locations, which vastly increases diversity and access; significantly reducing the time requirement, the travel expenses, and hence carbon footprint. This research truly leveraged this scalability property of GMB given the geographic diversity of the participants. However, some synergies of face-to-face interactions may be foregone.

Vennix [10] argued that the choice of the structure depended on the specific circumstances of a group, type of problem, the number of participants and their geographic locations and hence the processes could be tailored to specific situations. The method

for this research was tailormade due to the geographic and organisational diversity of the stakeholders.

### 2.2. Stakeholders Identification

GMB process starts with the identification of stakeholders to be involved. Mitchell et al.'s [11] broad dynamic typology of stakeholders for corporates was adapted here to determine not only a list but also a categorisation of stakeholders for the GMB. The approach proposes that classes of stakeholders can be identified by their possession or attributed possession of one, two, or all three of the following: power, legitimacy, and claim. This methodology supports managers identifying stakeholder salience in a dynamic manner and helps them prioritise stakeholder relationships.

Mitchell et al. [11] made three propositions: (1) stakeholder salience would be low where only one of the stakeholder attributes—power, legitimacy, and urgency—was perceived by managers to be present: dormant, discretionary, and demanding stakeholders fell into this category; (2) expectant stakeholders with moderate salience were perceived to possess two of the stakeholder attributes. Dominant, dependent, and dangerous stakeholders were prominent members of this group; (3) the definitive stakeholders had high salience as they had all three attributes.

However, on account of all the institutions involved in the EU fisheries' management as well as the other concerned parties the term "stakeholder" had to be more inclusive for the fisheries sector in order to incorporate all the various interested parties and social actors [12]. Combining Mitchell et al.'s [11] methodology with Newton et al.'s [12] inclusiveness criteria alongside Semeoshenkova et al.'s [13] marine-specific stakeholder definition resulted in the list in Table 1. The number and typology of stakeholders involved in the present study also met the criteria set out by Vennix [10] in terms of generating diversity of views to ensure the problem was properly defined.

**Table 1.** Classification of marine stakeholders.

| Institutions | Stakeholder Typology | Stakeholder Numbers Involved in the GMB |
|---|---|---|
| Fisheries managers | Definitive | 1 |
| Scientific advice bodies | Demanding | 3 |
| Researchers | Demanding | 4 |
| NGOs | Demanding | 4 |
| Fishing industry | Dependent | 2 |

Indeed, fisheries managers have all the three attributes: the power of taking decisions, the legitimacy of doing it and, of course, the urgency, especially when considering that in most cases deadlines (e.g., achieving MSY) have often been missed. The scientific world (advisors, pure researchers, and NGOs) has lower salience as they possess just one attribute: the urgency, as they act in defence of the environment or of the wider ecosystem's interests. They do not have the power to decide and their legitimacy may not always be fully recognised by the industry. Finally, the fishing industry clearly depends on other stakeholders (mainly upon policymakers), as most fisheries operate under regulations and permissions.

### 2.3. A Novel Approach to Participatory Modelling

The 14 stakeholders were provided with a preliminary CLD of the EU fisheries system. They were asked to reply to two questions to ascertain how their perception of the elements (variables) and causal links between these, in essence the state of the system, might differ from the one the preliminary model illustrated. Questions were:

- Do you agree with the typology and sign/direction of the identified relationships? If no, please, briefly specify.
- Do you believe we should include other variables and/or new connections among existing variables?

The model in Figure 2 was inspired by Dudley's [2] formative work of fisheries management but adapted to the specificities of EU fisheries after detailed literature review [2,7,8,14–19]. The key difference between Dudley's [2] model and this one is that the former assumed management's main target was to protect fish biomass, whereas the CFP mandate is much broader and encompasses all three aspects of sustainability. This bias towards maintaining a healthy fish stock in Dudley's [2] work was apparent from the crucial "perception of fish stock" variable and its connection to the desired vessel entry rate in the initial diagram as the most important management lever.

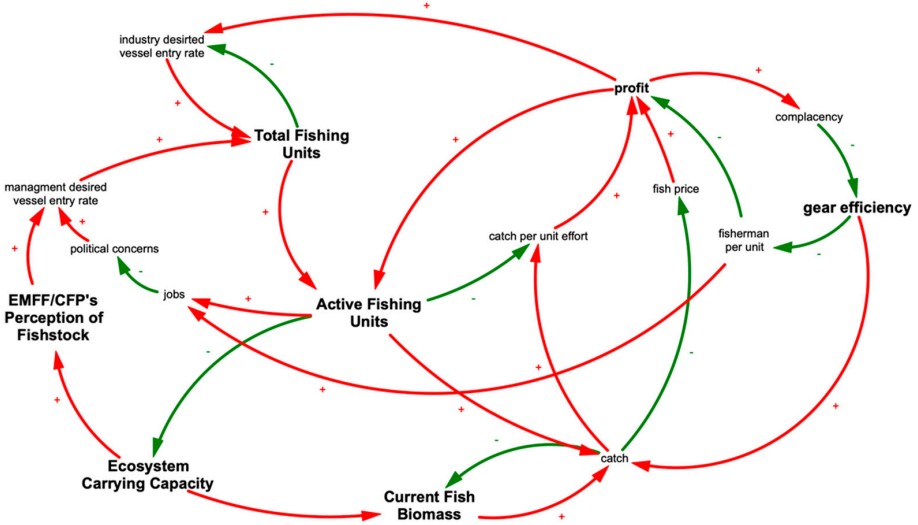

**Figure 2.** Preliminary causal loop diagram for group model-building process produced by Vensim PLE software.

The polarity of a causal relationship is illustrated with plus or minus at the arrowheads. A plus sign indicates that an increase in the independent variable causes the dependent variable to increase, *ceteris paribus* (and *vice versa*) as can be expressed as $X \rightarrow +Y \Leftrightarrow \partial Y/\partial X > 0$. On the other hand, when an increase in the independent variable causes the dependent variable to decrease, a minus sign is used, which can be expressed as $X \rightarrow -Y \Leftrightarrow \partial Y/\partial X < 0$ [20]. To augment polarity signs and enhance the visual appreciation of these relationships this research uses red arrows for reinforcing relationships and green arrows for balancing ones. Reasoning through the relationships by following the green and red lines it can be discovered whether a loop is a balancing or a reinforcing one. These positive (reinforcing) and negative (balancing) feedback loops are fundamental constituents of system dynamics [21] and endogeneity is one of the important traits of the SD models.

All correspondence was conducted through email, which proved to be an effective way of keeping the discussions progressing. The time between receiving the feedback emails from the participants and replying to them was essential for perusal of every input. The research was designed as a discreet process since the authors believed a continuous process of explicitly making revisions could have potentially led to biases of initial respondents steering the discussions. Thus, the revised model was shared once all feedback was reflected.

Feedback was very strong in the first round and, with two-way communications remaining open throughout this period, significant parts of the system structure and the issues to be addressed emerged. In the second round the discussion moved from

agreeing on the state of the system to the decision rules and hence comments centred around the management side of fisheries, as intended. However, participation moderated as management is a more specialised domain and some stakeholders preferred to follow the changes rather than offer their input.

The third round was conducted as a Zoom meeting where the final draft diagram was made available to the attending stakeholders to encourage participation. As the final CLD illustrates in Figure 4, the management of EU fisheries had to be divided into two broad regions, Mediterranean and Black Sea (MBS) and non-MBS. Given different socio-ecological characteristics of these regions in terms of ecosystems and nature of fisheries, input versus output measures were deployed, respectively, to effectively manage these areas. Fish populations tend to be smaller in the Mediterranean, supporting relatively smaller-scale, multispecies and multi-gear fisheries in a more fragmented sector [22]. In the interest of time only the MBS system dynamics stock-flow quantitative diagram was disclosed, which increased the conviction of participants in the methodology as abstract concepts in the CLD materialised into solid quantitative simulations with actual numbers.

The data used in the models, except for carrying capacity and initial fish biomass, are from the EC Report on the assessment of balance indicators for key fleet segments to achieve balance between fleet capacity and fishing opportunities [23], and Stella Architecture software was used to run the simulations. The models cover the 2008–2018 period, and the software continues final trends into 2050, unless an entered formula dictates otherwise. The policy decisions taken earlier are already reflected in the models, with their intended and perhaps unintended consequences.

## 3. Results

Although the qualitative and quantitative models were produced in conjunction with each other this section discusses them separately, since the quantitative scenario analysis were undertaken once the final CLD was agreed on by the stakeholders.

As anticipated, the participants suggested significant changes in the first round. The "Complacency" variable, included to consider the diminishing return of profits' impact on catching gear efficiency, was overruled and a direct link between profits and efficiency was established as the harvesting of natural resources is different from in other sectors. Catch per unit effort (CPUE), is a key determinant of catching gear efficiency. The stakeholders preferred to leave this relationship via profits. Consequently, the quantitative stock-flow model also used this structure to calculate changes in catching gear efficiency proxied as vessel power in the manner Palomares et al. [24] argued. Stakeholders rightly challenged "management desired entry rate", as one of the mandates of the CFP was to reduce the EU fishing fleet. However, the removal of this was left to the next review. Fishing units were divided into active and inactive with positive connections to employment. Ecosystem carrying capacity (ECC) had mixed feedback. Some argued that it was conceptual and should not be included. Other participants wanted to keep it and add a distinction between catch and juvenile catch as the latter had an impact on ECC. The impact of fishing on ECC was not considered. Despite its abstract nature ECC was kept since it was an essential variable for the quantitative stock-flows model to derive fish biomass numbers. Fish prices, both local and global, were added to allow for future scenario analysis. Management appeared with three different tools: catch quotas, landing obligations, and effort, derived from activity and active fishing units.

Overall, the revised version of the diagram as shown in Figure 3 was significantly more complex than the first version. This was a natural consequence of including almost all participants' feedback. Rationalising some of variables without any causal connections such as discards and exogenous factors were left to be addressed in the final round.

From second to the final round the most important amendment was the rationalisation of management measures for different regions. The first relates to the MBS region, where the control of fishing effort is the main management device, alongside the use of technical measures. The second and the third channels are for non-MBS regions and the measures

are catch quotas, technical measures, landing obligations, and area closures. The EMFF was omitted since the institution is only a funding vehicle for the CFP. Landings substituted catch as the former are actual numbers and the latter include estimated discards. Discards were omitted since they did not have a causal connection with any other element, and they were not operational at this juncture due to de minimis exemptions in the MBS regions. The bulk of the MBS fleet is composed of small-scale vessels (smaller than 12 m in length) and de minimis are extensively used in order to comply with the requirements of the landing obligation. Inactive vessels and other related activities were removed as they did not change the behaviour of the model. Management-desired vessel entry rate was omitted considering the cap on new licencing and data supported a continued slide in vessel numbers in both regions. Resource rent was replaced by net profit due to terminology issues. Fish prices were left due to stakeholder insistence, but they had no impact on the model's behaviour.

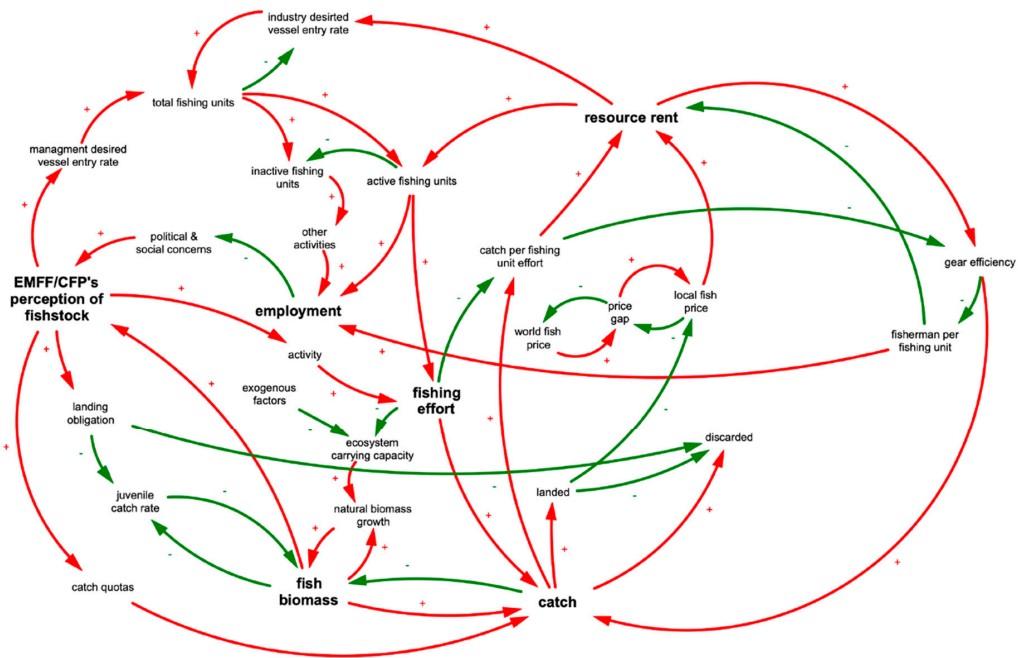

**Figure 3.** Revised causal loop diagram produced using Vensim PLE software with stakeholders' input and other suggested research [25–31].

As the final step of the methodology, stakeholders were presented the evolution of the CLD in the first two stages to reach the final model in a live Zoom meeting. There was broad consensus on the final CLD, and no other changes were proposed by the participants during the call. The quantitative stock-flow diagram for the MBS region that aided the formulation of the CLD was also shared with the participants as proof of concept. Seeing the model operational increased their conviction in the methodology. The final version in Figure 4, with a summary of the changes, was emailed to all stakeholders, which brought the participatory modelling process to an end.

The completion of the qualitative model and significant buy-in by the participants increased the conviction in the results of the stock-flow diagrams co-generated to support the GMB. The two quantitative models showed that it was indeed possible to achieve CFP's mandate of addressing all three aspects of sustainability with the right set of policies. These two models, for two respective regions, were also made available online to stakeholders to allow them to conduct their own scenario analysis; to aid policy formulation; and to mitigate the chances of unintended consequences of policy decisions.

In these stock-flow models three aspects of sustainability were proxied as follows: fish biomass represented the ecological, engaged crew the social, and profits the economic pillars of sustainability. Starting with the MBS region, there has been a clear downward

trend in the number of fishing vessels as a direct consequence of the CFP policy to control inputs (primarily based on limiting fishing effort and improving gear selectivity) as the main management tool to balance fishing capacity with fishing opportunities. Although fishing days are still used selectively as a policy tool these were left unchanged in this model since they were already low. Therefore, the key policy lever remains the vessel number, where the well-established downward trajectory has had a positive impact on profits as costs come down. Allowing the current downward trend to extend into the future increases investment due to higher profits, which leads to higher gear efficiency and higher landing per unit effort (LPUE), albeit up to a point, which is incorporated through a diminishing return behavioural hypothesis, which suggests there is a limit to LPUE gains from gear efficiency and it will tail off over time. Profits remain stable and fish biomass recovers gradually on the back of lower fishing effort. This process fulfils two out of the three pillars (ecological and economic). However, this loop has the adverse effect of lowering employment and therefore challenging the social aspect of sustainability (Figure 5).

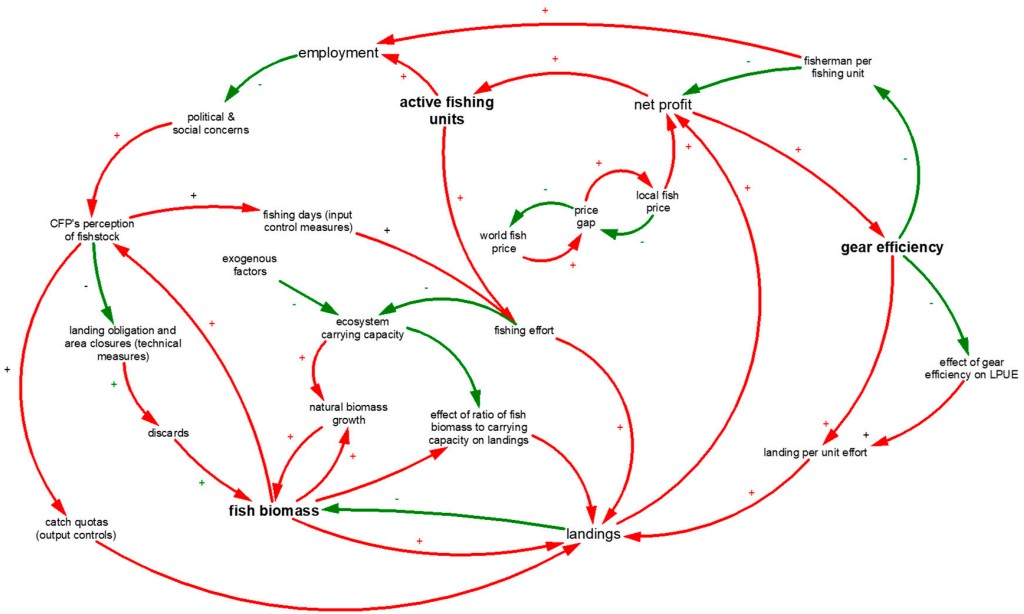

**Figure 4.** Final causal loop diagram produced using Vensim PLE software.

Consequently, recalling the "political concerns" of the CLD, intervention to cease this reduction in vessel numbers seems inevitable from a policy point of view. Applying a minimum number of 12,000 vessels for the MBS region as a whole, for example, reduces the employment losses with some delays, but 16,000 alleviates this problem as illustrated by the green and red lines, respectively. However, if the floor is set at higher levels the fish biomass growth comes down significantly. This scenario perhaps dispels the criticism raised about the cap applied to the entry of new fishing vessels and limits to improvement in fishing gear for this particular region.

For the non-MBS region, Figure 6 shows three different scenarios. Scenario one is the current trajectory (blue lines) with an assumption of 100 mn tonnes of quota increase per annum. Although output measures such as catch quotas are used as a management tool, reduction in vessel numbers, similar to the MBS region, outweighs TAC restrictions. Both ecological and economic sustainability appear to be achieved but sustaining the welfare of fisher-people requires other policy measures. In scenario two, as per the MBS region, a floor of 30,000 is applied to vessel numbers to arrest the falling engaged crew numbers. However, compounded by 100 mn tonnes of quota increase per annum this leads to collapse of the fish biomass in the later years (please note that red lines are not visible in the first two charts as they overlap with the next scenario output). To address this undesired outcome, quota increase is reduced by half to 50 mn tonnes per annum while minimum vessel numbers is

kept at 30,000 in the final simulation, which turns out to be the optimum scenario in terms of addressing all three aspects of sustainability.

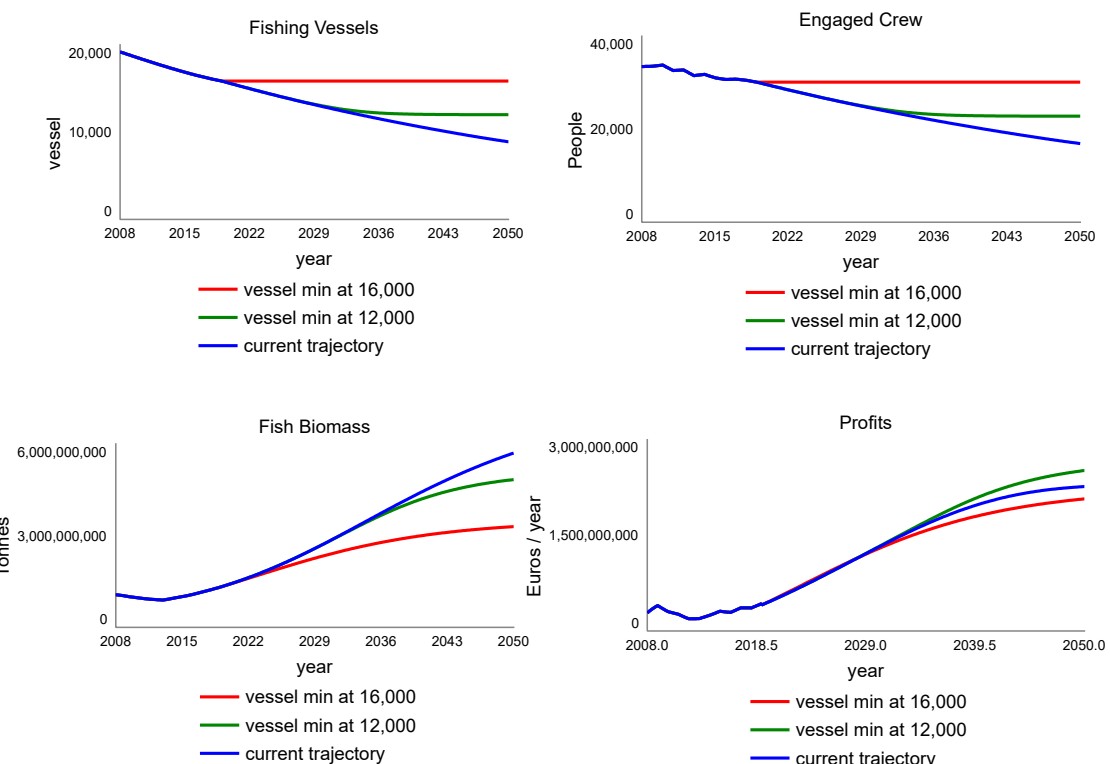

**Figure 5.** MBS region policy simulation graphs produced using Stella Architect software.

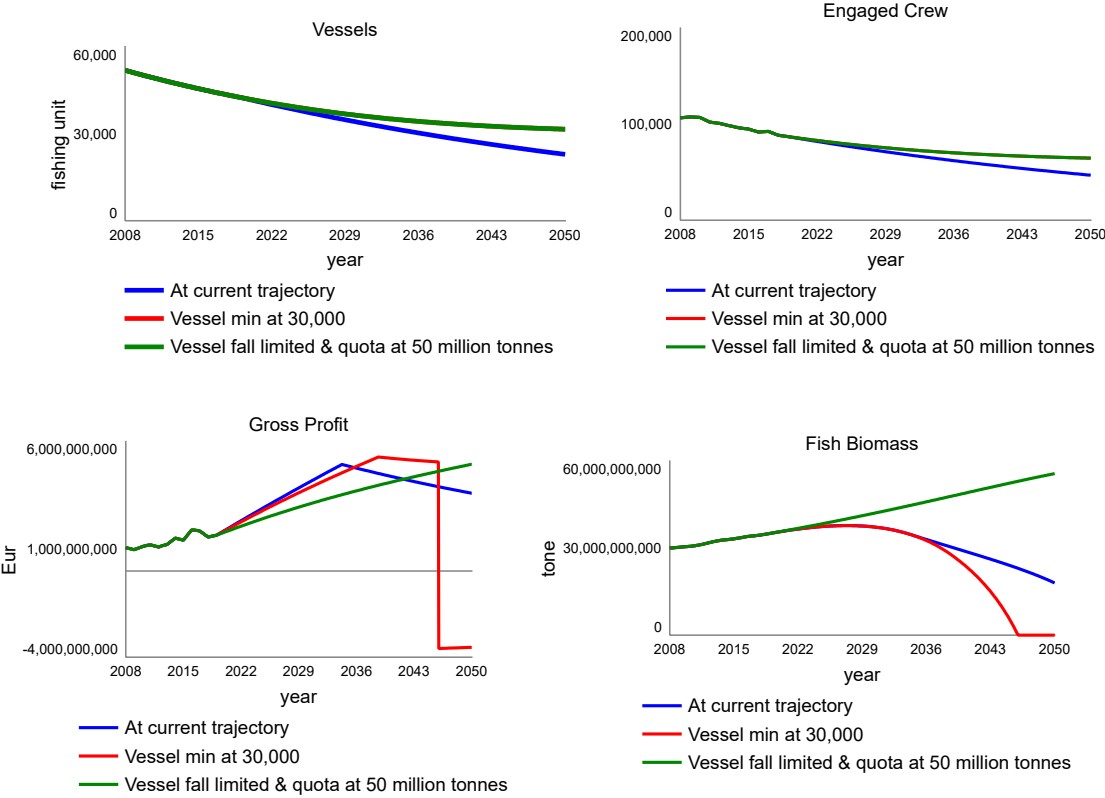

**Figure 6.** Non-MBS region policy simulation graphs produced using Stella Architect software.

The key challenge was the absence of reliable biomass numbers. Thus, the model was calibrated on actual landing numbers. Carrying capacity and initial fish biomass numbers were estimates that were approximated during the calibration phase of the models so that the actual historic numbers for all other variables concurred with the model generated data. These two variables were included in the CLD, despite some stakeholder views of them being conceptual, as they were crucial to construction of the stock-flow model.

The models are valid with robust Internal structures [30], tested by comparing empirical evidence with the model output and gradually building confidence that the models are producing the right output for the right reasons [31]. Barlas [30] argued that validity could also be increased by introducing new variables with new interrelationships and increasing the system boundaries could help better understand the system behaviour. In the second round of the GMB process new variables and their connections were introduced by the participants, making the CLD more complex. However, in the last round, the stakeholders eliminated those variables that did not help explain the behaviour of the system and agreed on a final model. This part of the process was particularly important since Forrester and Senge [31] further claimed that validity in the SD models could be increased by gaining the confidence of people that were not directly involved in the construction.

## 4. Discussions

The novelty of this research was threefold. First of all, it was designed and implemented as a fully online group modelling. This consequently helps future researchers better calibrate the number and diversity of participants necessary to model a complex system with management challenges and the number of researchers required to conduct such research, the latter firmly depending on the choice between synchronous and asynchronous process. There are differences between the GMB mentioned in research articles [32–34] and the one undertaken for this research. For example, to construct a model Vennix [10] suggested nine to fifteen participants in a face-to-face conference room environment with a range of views to assure that the problem is properly defined and avoid the risk of groupthink. At the time this research was undertaken there was no precedence as to the number of stakeholders necessary for an online GMB. The first such paper was published by Wilkerson et al. [35] in late 2020, as COVID-19 affected travel plans, propelling the researchers to fully imitate a face-to-face GMB online with a homogenous group (number not specified) of Norwegian nationals.

In terms of the number of researchers, face-to-face model building often requires a facilitator, recorder, modeller, process coach, and gatekeeper. Vennix [10] mentioned two to five modellers to facilitate the live process. Wilkerson et al.'s [35] information elicitation phase was synchronous. Although their research does not make it clear, their presentation for the System Dynamics Society gave the impression that all six co-authors were actively involved in the GMB process in various roles. One of the key advantages of the asynchronous modelling process was that a facilitator, Nisea, and a modeller were enough to conduct this modelling process. The other benefit of the asynchronous approach was that it gave the researchers time to carefully reflect on the comments, study the articles, and analyse the data to better appreciate the reasoning behind suggested revisions. The facilitator was also instrumental in making sure that every stakeholder was on board at the synchronous third and final round.

The second novelty is that the initial mandate of generating a qualitative participatory model was aided by two quantitative stock-flow diagrams with actual numbers, which help deepen the understanding of the sustainability trade-offs inherent in the complex fisheries system for all stakeholders and give the policy makers a tool to properly and holistically calibrate policy decisions.

However, the main disadvantage of the online process could potentially be that some synergies gained from face-to-face interactions might have been lost. One of the key strategies of the preferred research method was to avoid group-thinking by reflecting each individual input in the model in turn and making the revised model available once all inputs were included. Although some of the stakeholders knew each other, there was no

evidence of collaboration. That said, given the geographic diversity of the stakeholders, to bring them to one location to undertake this modelling process could have resulted in excessive financial and manhour costs and led to an unnecessary increase in carbon footprint, especially during the COVID-19 pandemic.

Vensim PLE software from Ventena Systems Inc. was used as the CLD model building tool to generate the first draft of the model shown in Figure 2. This diagram was shared with the stakeholders via email alongside a letter explaining causal loop modelling as the qualitative SD tool. A choice had to be made early in the process between providing a preliminary model versus allowing stakeholders to determine the variables and their relationships. Vennix [10] argued the pros and cons of both methodologies with no clear favourite. In their research Haapasaari et al. [36] discovered that stakeholders were sceptical about their ability to identify fishery variables and their linkages due to epistemic and variability uncertainty. Otto et al. [37] conducted GMB for fisheries as well, looking at solving the sustainability problems faced by the Gloucester Community Development Cooperation in Gloucester, MA in the USA. Their approach was different from both this and Wilkerson et al.'s [35] research insofar as they did not get their client involved in model building. Their client provided data and operational details of a proposed factory. They admitted to having a tough time building confidence in the model but eventually got there as they claimed that the model "had a strong organizing and internalizing impact" [37] (p. 310). Given the hybrid process planned for this research, offering an initial model with a preliminary number of variables and their relationships in a CLD format was preferred for facilitating and streamlining the discussions.

## 5. Conclusions, Limitations and Further Research

This research applied system dynamics methodology to fishery management through online group model-building with the participation of carefully selected diverse set of stakeholders. This transparent and participatory process not only elicited the knowledge of many stakeholders to model the system but also fulfilled learning by modelling and buy-in with participation criteria of the GMB process. Finally, it offers managers a toolbox to better calibrate policy choices before they are implemented to mitigate unwanted outcomes.

As per the limitations, with longer historic data, calibration phase of the stock-flow model building process could have been easier and have increased the conviction in the robustness of the outcomes. In particular, modelling the MBS region was a formidable task, partly caused by the exclusion of two significant fishing countries, Greece and Croatia, due to missing data points. Additionally, the biomass numbers remain the most important challenge to the stock-flow diagrams and new ways to quantify them can help researchers better understand and calibrate SD models such as this one. Thus, system dynamics models for fisheries could be more potent when applied to single fisheries and areas, including selected fleet segments exploiting selected fish stocks. Finally, future researchers can undertake face-to-face group modelling of fisheries with a similar set of diverse stakeholders and methodology so that the outcomes could be compared to understand if the online version envisaged in this research lost any important synergies.

**Author Contributions:** Resources, M.G.; Data curation, L.M.; Writing—original draft, E.G. All authors have read and agreed to the published version of the manuscript.

**Funding:** This research received no external funding.

**Informed Consent Statement:** Not applicable.

**Data Availability Statement:** European Environment Agency, (n.d.) https://www.eea.europa.eu/data-and-maps/indicators/status-of-marine-fish-stocks-4/assessment; https://stecf.jrc.ec.europa.eu/reports/balance/-/asset_publisher/3rBi/document/id/2789763?inheritRedirect=false&redirect=https%3A%2F%2Fstecf.jrc.ec.europa.eu%2Freports%2Fbalance%3Fp_p_id%3D101_INSTANCE_3rBi%26p_p_lifecycle%3D0%26p_p_state%3Dnormal%26p_p_mode%3Dview%26p_p_col_id%3Dcolumn-2%26p_p_col_pos%3D1%26p_p_col_count%3D2 (accessed on 18 September 2022).

**Conflicts of Interest:** The authors declare no conflict of interest.

**Appendix A**

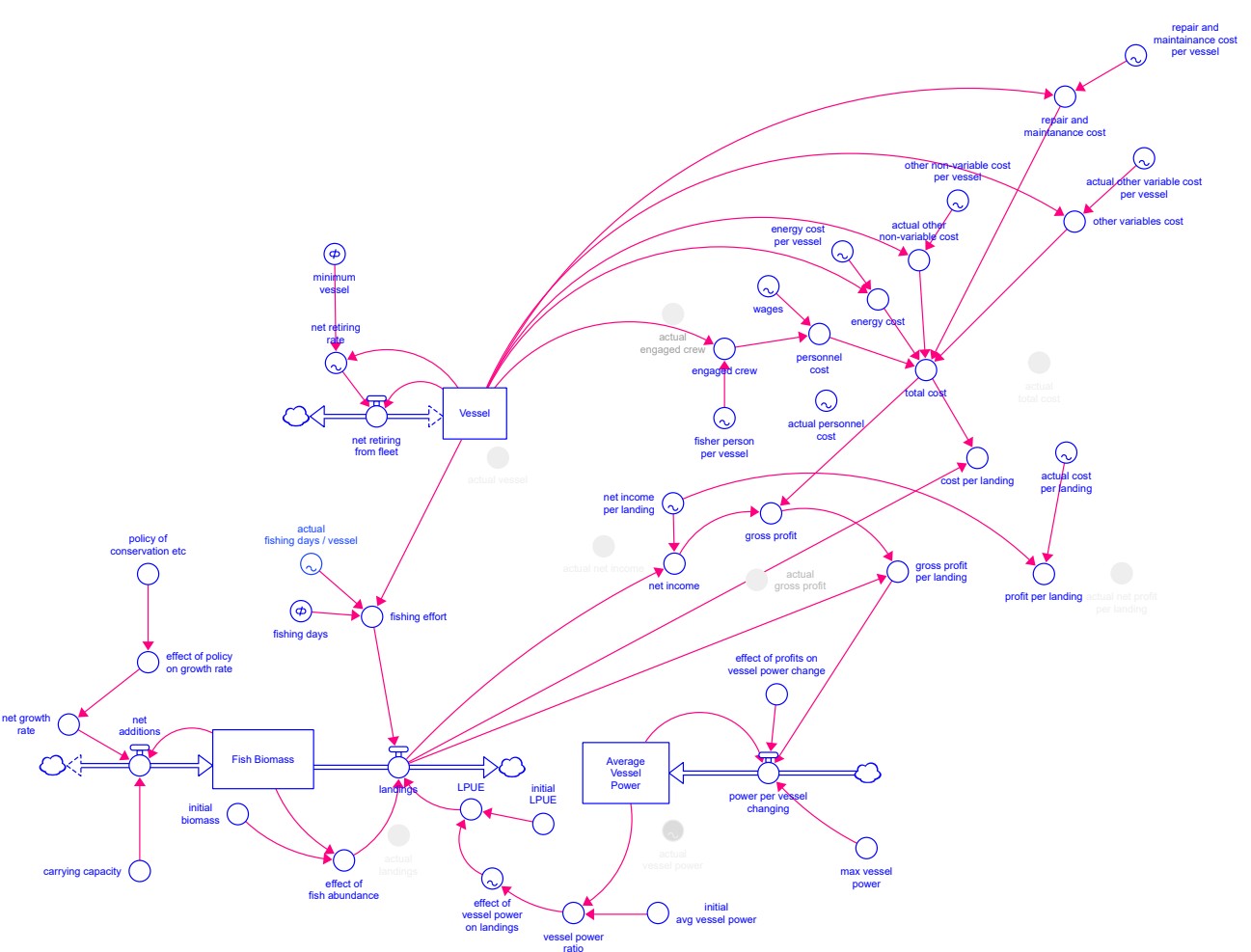

**Figure A1.** Stella Architect-generated stock-flow model for MBS region.

The interactive model and the story can be reached at https://exchange.iseesystems.com/public/erda/mbs-model.

The interactive model and the story can be reached at https://exchange.iseesystems.com/public/erda/non-mbs-model.

Stella Architect-generated stock-flow model equations for both regions can be reached at https://drive.google.com/file/d/1adV9cI-I47-r1e6Pi1hPXjYQVYTxnic5/view?usp=share_link.

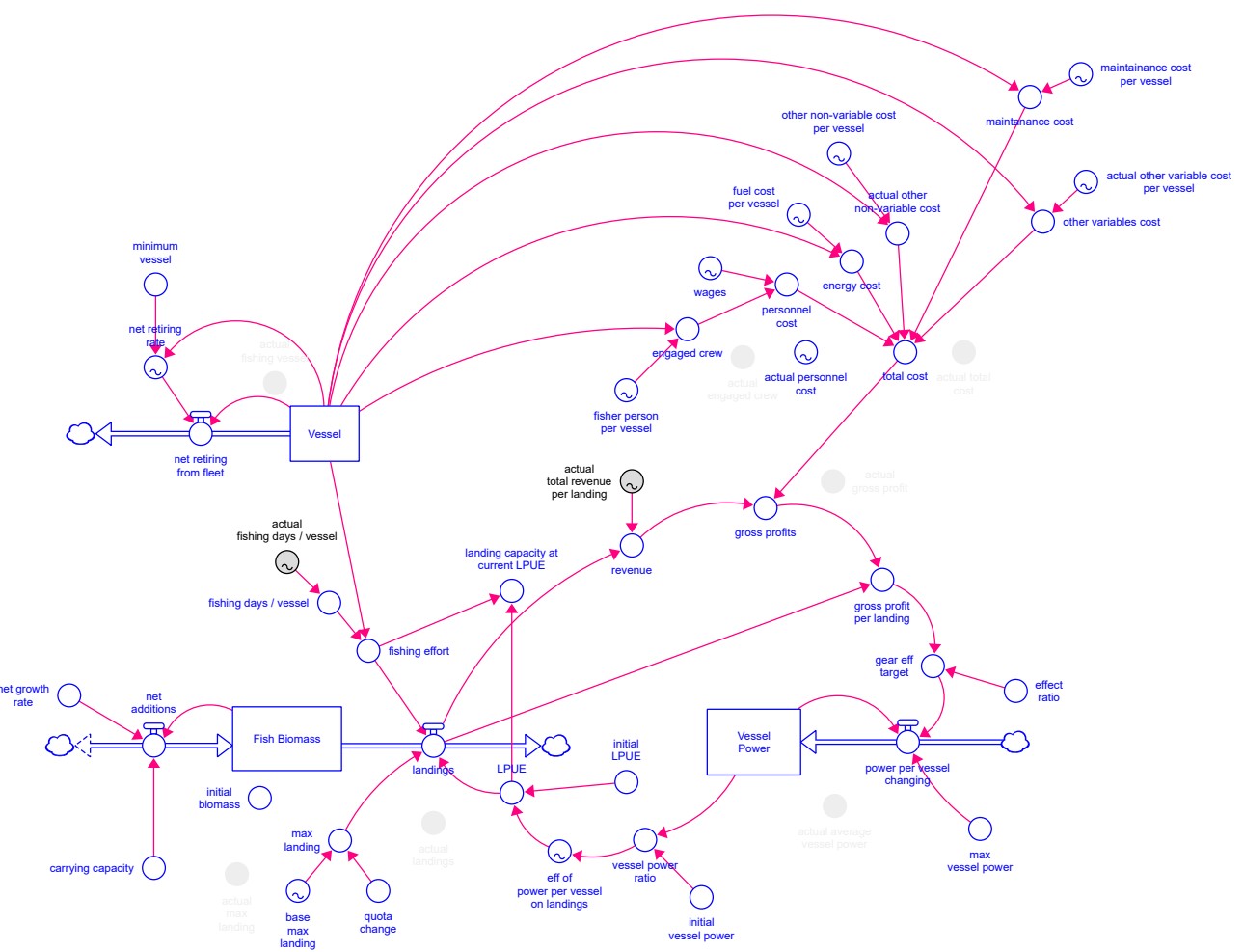

**Figure A2.** Stella Architect-generated stock-flow model for non-MBS region.

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
