# Peer review of "Understanding EU Fisheries Management Dynamics by Engaging Stakeholders through Online Group Model-Building"

_sustainability, doi:10.3390/su142315862_

Round 1
Reviewer 1 Report (New Reviewer)
General Remarks:
The paper applied system dynamics methodology to fishery management through online Group Model Building.
However, major revisions of the manuscript are needed for publishing the paper.
The paper is not yet ready for publication. Major revisions are required
Specific Remarks:
1. Abstract: The abstract should state briefly the major conclusions. The abstract is often presented separately from the article, so it must be able to stand alone.
2. The introduction can be improved. The authors should explain: WHY Mediterranean and Black Sea (FAO fishing area 37)?
3. Materials and Methods: The authors should provide sufficient detail regarding materials and methods: to allow the work to be reproduced by an independent researcher.
4. Results: What is the validity of the model?
5. The limitations of this paper should be added at the end of Conclusions.
6. In my view the wording is sometimes misleading, ambiguous, and many arguments are not well enough supported by references.
7. “The quantitative stock-flow model also used this structure to calculate changes in catching gear efficiency proxied as vessel power” WHY?
8. Why the impact of fishing on ECC was not considered? (page 8).
Author Response
Abstract: The abstract should state briefly the major conclusions. The abstract is often presented separately from the article, so it must be able to stand alone. Abstract has indeed the conclusion: Final model evolved significantly from the initial one offered, which pointed to active involvement in and progressive learning from the modelling process itself, as the methodology argues. Two quantitative stock-flow models using actual numbers were built not only to aid the GMB process but to depict how all three aspects of sustainability could actually be met with the right set of policies that consider feedback loops and inherent trade-offs.
- The introduction can be improved. The authors should explain: WHY Mediterranean and Black Sea (FAO fishing area 37)?This is already explained on page 7
- Materials and Methods: The authors should provide sufficient detail regarding materials and methods: to allow the work to be reproduced by an independent researcher. In the last round, we were told to concentrate on our work. We restructured the manuscript to emphasise our research, still keeping enough details for the work to be reproduced. CLD’s can be instantly replicated and expanded by future researchers. Quantitative models as well as model equations are made available online and can be accessed through links provided with the models in the Appendix.
- Results: What is the validity of the model? Added: The models are valid with robust internal structures [27], tested by comparing empirical evidence with the model output and gradually building confidence that the models are producing the right output for the right reasons [28]. Barlas [27] also argued that validity could also be increased by introducing new variables with new interrelationships and increasing the system boundaries could help better understand the system behaviour. In the second round of the GMB process new variables and their connections were introduced by the participants making the CLD more complex. However, in the last round the stakeholders eliminated those variables that did not help explain the behaviour of the system and agreed on a final model. This part of the process was particularly important since Forrester and Senge [28] further claimed that validity in the SD models could be increased by getting the confidence of people that were not directly involved in the construction.
- The limitations of this paper should be added at the end of Conclusions. Limitations were already on page 13, the last paragraph of the text.
- In my view the wording is sometimes misleading, ambiguous, and many arguments are not well enough supported by references.
- “The quantitative stock-flow model also used this structure to calculate changes in catching gear efficiency proxied as vessel power” WHY?Horsepower is considered a measure of the effective measure of effort. Added: Palomares, M. L. D., and D. Pauly. 2019. On the creeping increase of vessels’ fishing power. Ecology and Society 24(3):31. https://doi.org/10.5751/ES-11136-240331
- Why the impact of fishing on ECC was not considered? (page 8).This requires indicators that are not available region-wide. The limitations section also mentions this issue.
Reviewer 2 Report (New Reviewer)
Manuscript reference: Sustainability-1951403
Manuscript title: Understanding EU fisheries management dynamics by engaging stakeholders through online Group Model Building
Review report
Date: 9th September 2022
Overview and general recommendation
Development and implementation of an effective and sustainable fisheries management policy has been a challenge all over the world. This is demonstrated by the continuous declining of fish stocks. One of the major cause of the failure in fisheries management policies is the lack of an adequate stakeholder involvement in the management policy development and implementation.
The stakeholder involvement in policy development and implementation is important because it tries to bring the relevant stockholders together, understanding and paying attention to what is important to each and every stakeholder, identify the individual and common issues, which can foster connections, trust, confidence, and buy-in, and commitment for the implementation of the policy.
However, where there is a large number of stakeholder, with different interest and opinions, it turns difficult to assure an effective stakeholder alignment. Hence, the present study, applied the Group Model Building (GMB) participatory approach tool, combining qualitative and quantitative approaches for stakeholder engagement in the complex issues of EU fisheries management.
The study demonstrated the need and relevance of an adequate engagement of the stakeholders in fisheries management development and implementation, and the challenges involved for stakeholder engagement. Online stakeholder consultation proved to be effective for a wider stakeholder engagement. Hence, the study is very relevant for both scientists and managers.
The authors have applied correct methodology and consulted relevant literature. The manuscript was revised for English language and grammar. Indeed, the manuscript is well structured and well written. Therefore, I recommend the manuscript to be accepted as it is.
Author Response
Thank you for your kind review.
Reviewer 3 Report (New Reviewer)
Dear author(s)
The Common Fisheries Policy in the EU has a significant impact on the various fisheries relationships in EU Member States. In order to predict the impact of this policy on these stakeholders in advance, it is necessary to identify their interdependencies. This study responds to this requirement. The study has achieved some results by utilising the Group Model Building Approach. However, the study still has several areas for improvement.
The first is to conduct a survey of previous studies. The introduction lacks a subsection or paragraph referring to previous research. This makes it unclear why this study should be conducted and its significance.
Second, it cites studies that are relevant to this analytical approach. This citation may reinforce the interdependencies proposed by this study. See the papers listed in the references section.
Third, the table on page 5, line 212, for example, is not located in the center of the manuscript. Hence, it should be corrected so that the table is placed in the center of the manuscript. Authors should probably recheck the formatting of the manuscript.
References
[1] Hamaguchi, Y. (2022). Welfare effect of rent-seeking activities under international management of fishery resources. Fisheries Research, 246, 106170.
[2] Utami, T. N., Fattah, M., & Iintyas, C. A. (2022). The System Dynamic of Mangrove Ecotourism of “Kampung Blekok” Situbondo East Java Indonesia: Economic and Ecological Dimension. Environmental Research, Engineering and Management, 78(2), 58-72.
Author Response
Thank you for your review and feedback.
On the first point, the sub-sections were removed by the previous reviewers.
On the second point, again previous reviewers asked us to reduce the number of citations and concentrate on our work, the research we conducted. We appreciate the research that you recommended. After perusing the document we decided that they would not help enrich our research.
On the formatting issue, the editor seems to have applied their own formating, which resulted in some figures being misaligned. All these reviewed and corrected.
Round 2
Reviewer 1 Report (New Reviewer)
I have checked the revised version. Overall Recommendation: Accept in present form.
Reviewer 3 Report (New Reviewer)
Dear Author.
This manuscript has been carefully revised in accordance with a number of reviewer comments. This means that the manuscript has reached the standard of this journal. I would therefore like to recommend to the Editor-in-Chief to accept this manuscript in this journal. Good luck.
This manuscript is a resubmission of an earlier submission. The following is a list of the peer review reports and author responses from that submission.
Round 1
Reviewer 1 Report
Understanding EU fisheries management dynamics by engaging stakeholders through online Group Model Building needs substantial edits, reformatting, and reworking. Overall, the authors need to work on concise scientific writing. The first half of the abstract should be reduced to 1-2 lines of background. Overall, please focus on giving a concise background a reader needs to know to focus on your work. The entire introduction reads like a term paper for a course. It should be reduced to 4-5 total paragraphs. Instead of giving detailed quotes of many citations, the important thoughts and findings relevant to an introduction should be included. Many articles like The and A are missing. The introduction should only include figures that are new and necessary. All three in the introduction should be deleted.
Your objective is buried in lines 56-60. Your introduction should end with your objective for this research or purpose. There should also not be the need for subheading in the introduction.
The citations are not in the proper format with numbering, and the bibliography should be references. Additionally, the entire reference section is in the wrong format. I encourage the authors to review some other published papers in the journal as well as read the author’s guide. A citation style can be downloaded for most common citation managers.
All of 2.1-most of 2.4 fits into introduction. The Materials and Methods (note wording) should focus on your methods with citations used when summarizing previously published methods. Line 385- missing first half of the sentence. Line 391-418- not methods. Delete Fig. 6 Methods should be past tense.
The results should not just be bulleted points. Are only 15 stakeholders the total sample size used in the project? If so, this might be useful as a pilot, but no real results can be drawn with only 1 manager and 2 of most other categories. Line 556- rightly argued…judgment should be kept out of the writing.
Reviewer 2 Report
The manuscript titled “Understanding EU fisheries management dynamics by engaging stakeholders through online Group Model Building” submitted for publication in Sustainability presents a research tool that aims to offer a new methodology for a participatory model building process enabling key stakeholders to get intensely involved in identification and modelling of the fisheries management problems and solutions. A Group Model Building (GMB) tool of System Dynamic methodology is used to assembly the views of diverse groups of stakeholders to generate a commonly agreed fisheries causal loop diagram (CLD).
Certainly, a strong and constructive stakeholder involvement is a key issue in modern fisheries management. That has been clear at least the last twenty years. All efforts and approaches to improve the level of involvement and understanding of its challenges are highly welcome. According to the authors of the submitted manuscript, the research on the subject lacks a holistic view of the entire system to underpin policy decisions, and that prevents the reliable anticipation of the consequences of these decisions. The authors state that the tool they present is novel and holistic and allows a universal understanding of the system by a adding a systematic input from stakeholders. The authors note that their study is unique and there is no other similar study to compare their findings. No critical assessment of the potential sources of errors was made.
In essence, most the issues these authors raise have been under intensive research for several decades. There are thousands of studies where all the three pillars of sustainability have been under investigation; the current study certainly is not the only one although the approach may be new. The authors should make clearer what exactly is novel in their study. The fact that the interviews and interactions were made remotely may not be that different from face-to-face interactions.
It is well known that fisheries are complex systems and extremely difficult to manage. It has been demonstrated again and again that effective fisheries management is a necessary element in successful and sustainable fisheries. The authors should be more explicit what is novel in that respect in their study.
Although innovative through the methodological approach, the study unfortunately has many other features that require revision. First of all, the paper is written merely like a book chapter and is unnecessarily long. For instance, the introduction is almost 7 pages and most of text is general knowledge that can be easily found in literature and the web. The introduction should only state the objectives of the work and provide an adequate background, avoiding a detailed literature survey or a summary of the results of other studies. Clearly, the current introduction requires a substantial reduction and much more focus. There is no need to repeat well-known issues. Furthermore, the text of the manuscript in general lacks the clarity, and in many parts appears like a conceptual jungle. The text needs stringent editing, and the language needs to be improved.
Below are some specific observations where the authors might want to improve their presentation.
There is no need to stress in the abstract that FAO underlines the importance of fisheries management. It is a well-accepted fact and has been repeated several decades by many fisheries management organizations. The abstract should state briefly the purpose of the research, the principal results and major conclusions. It must be able to stand alone. Major revision in needed.
In the first paragraph of the introduction, the authors claim that FAO has its own fisheries monitoring program. That is not exactly true. The member countries and other international fisheries institutions (such as ICES) are conducting the monitoring. FAO merely just collects and processes that data.
In the same paragraph, the authors claim that in the Baltic Sea there has been signs of recovery in the reproductive capacity of several fish and shellfish stocks. That is not true. Please delete the “Baltic Sea” from the sentences.
I am not sure whether Figure 1 is necessary. It can be easily found in the web.
The chapter “Methodology” on pages 8-15 appears pretty much similar as the introduction. It apparently aims to give the background for this study. It is way too general and way too long. Material and methods should provide sufficient details to allow the work to be reproduced by other researchers. Earlier methods that are linked to the topic should only briefly be summarized and indicated by a reference. There is no need to explain in detail what other scientists have done. If quoting directly from a previously published method, use quotation marks and cite the source. A Theory section should not repeat the background to the article already dealt within the introduction. Major revisions needed. The authors really should focus on the work they have done, and they should do it in a brief, focused and concise manner. This is not a textbook chapter. Just write what you have done and nothing more.
The “Results” chapter requires substantial improvement. Results should be clear and concise. More clarity is needed, and some issues could be presented as small tables. Please explain what you mean by the word “Complacency”?
The “Discussion” chapter is long and winding. Discussion should explore the significance of the results of the work, not repeat them. Furthermore, please avoid extensive citations and discussion of published literature. Focus on your own observations. Please start with the most important finding of your study. Do not expect that the readers are aware of the studies of Dudley and other scientists you consider important, and do not explain what kind of gaps you may have filled in their studies. Just tell clearly what you have found in your study and what is the importance of those findings. As in the earlier chapters, also here the authors are repeating many well-known issues. That is not necessary.
Please explain what you mean by the wording “gear characteristics”?
Please explain what you mean by the wording “higher gear efficiency”?
Please explain what you mean by the wording “diminishing return behavioural hypothesis”?
Figure 10 and 11 suffer of small size and small font size. It is difficult to see the points. Furthermore, the text poorly supports the content of these figures. Requires improvement.
There should be a critical assessment of potential sources of errors in the approach and conclusions and perhaps some ideas of the foundation of the further work. It would have interesting to have a bit more discussion of the specific features of this study (remote interactions with stakeholders) versus more traditional face-to-face interactions. What were the benefits and disadvantages?
The main conclusions of the study could be presented in a short section. The current conclusion is way too long and appears more like an extension of the discussion. Major revision needed.